# Designing a Deep Q-Learning Model with Edge-Level Training for Multi-Level Task Offloading in Edge Computing Networks

Ahmad Zendebudi *[ID] and Salimur Choudhury [ID]

Department of Computer Science, Lakehead University, Thunder Bay, ON P7B 5E1, Canada
* Correspondence: azendebu@lakeheadu.ca

**Abstract:** Even though small portable devices are becoming increasingly more powerful in terms of processing power and power efficiency, there are still workloads that require more computational capacity than these devices offer. Examples of such workloads are real-time sensory input processing, video game streaming, and workloads relating to IoT devices. Some of these workloads such as virtual reality, however, require very small latency; hence, the workload cannot be offloaded to a cloud service. To tackle this issue, edge devices, which are closer to the user, are used instead of cloud servers. In this study, we explore the problem of assigning tasks from mobile devices to edge devices in order to minimize the task response latency and the power consumption of mobile devices, as they have limited power capacity. A deep Q-learning model is used to handle the task offloading decision process in mobile and edge devices. This study has two main contributions. Firstly, training a deep Q-learning model in mobile devices is a computational burden for a mobile device; hence, a solution is proposed to move the computation to the connected edge devices. Secondly, a routing protocol is proposed to deliver task results to mobile devices when a mobile device connects to a new edge device and therefore is no longer connected to the edge device to which previous tasks were offloaded.

**Keywords:** mobile edge computing; task offloading; deep Q-learning; optimization

## 1. Introduction

Small, portable devices have limited processing and energy capacity, but they are usually required to perform processing intensive tasks such as image processing, augmented reality, real-time sensory input processing [1], and Internet of things [2]. An obvious solution to this issue is to use more advanced and power-efficient chips. This, however, is not always possible, as the use of cutting-edge chip technologies can be very expensive and might require many more years of advancement before it becomes a viable solution. Another approach is to offload tasks to the cloud, where the tasks are executed and the results sent back to the portable devices. This method also is not always possible, as some applications require a very small processing delay and transmitting data over the network through several routers is simply too slow to be applicable. A third method is to use a set of computational devices located at the edge of the network. That is, these computational devices are either directly connected to mobile devices or they are much closer to the mobile devices compared to cloud servers [3]. Using this approach, mobile devices can offload some of their tasks to these devices while maintaining a minimal transmission delay. As a result, mobile devices can meet the time-delay requirements of their tasks without requiring more advanced chips or running out of battery power [4].

By choosing the right tasks to offload to edge devices, edge processing can boast the advantages of both local execution and cloud processing, while avoiding the disadvantages associated with them. That is, selectively offloading tasks to edge devices can lower the processing time to much lower than what can be achieved using a cloud solution or local execution, as it avoids the latency of task propagation to the cloud or the slow execution of

tasks locally. Additionally, it can provide the same energy-saving advantages of a cloud solution without congesting the network or requiring more advanced and capable chips.

However, it is not always clear which tasks should be offloaded to edge devices and which ones should be executed locally. For instance, a task with a high computational requirements but a small task size might be more suitable for task offloading compared to a task with lower computational requirements and a large task size, as the transmission of a large file can be more time consuming than local execution of that task. Additionally, the decision can also depend on the current workload of the mobile and connected edge devices. If a connected edge device has a light workload while the mobile device has a large number of tasks queued for local execution, then it might be better to offload a task even if it has a large task size. One method for tackling this uncertainty is to hard-code offloading rules for a specific environment. Such an approach, however, will be too rigid and will require reconfiguration whenever the environment changes. Another approach is to use a reinforcement learning agent to make the offloading decisions. In this work, we use a deep Q-learning model in mobile and edge devices to make the offloading decisions. Training an artificial neural network for the Q function of the Q-learning model, however, can require significant computational and energy resources by itself, which is a burden for the limited computational and energy capacity of mobile devices. This is an issue that is barely addressed in the literature. In this work, we address this issue by training the neural network for the Q function in edge devices and sending the trained network to mobile devices for decision-making.

## 2. Related Work

The literature around task offloading varies considerably with regard to the network model, the optimization criteria, and the methods used for optimization.

In [5], all connections are wireless, even the ones between edge devices. Additionally, each mobile device is connected to at most one edge device, while edge devices can be connected to multiple edge devices. In this model, 100 mobile devices are randomly positioned in a plane, while four edge devices are positioned on four corners of a rectangle. Each edge device in this model can further offload a task to another edge device as long as the task was not received from an edge device. For the optimization criteria, they consider a combination of task response delay and energy consumption of mobile devices. Two methods are used for choosing the tasks to be offloaded: the mathematical programming and deep Q-learning [6] methods. It demonstrates that the deep Q-learning method can achieve results with similar performance as the mathematical programming solution with the advantage of being an online machine-learning method.

Karimi et al. in [7] propose a deep reinforcement model for vehicular mobile devices. In their work, a mobile device can have a number of applications that, in turn, produce tasks with different quality of service (QoS) requirements. An acceptance criteria is then introduced, which indicates whether a task can be accepted by an edge device considering its QoS requirements. For the optimization criteria, the aim of a solution is to maximize the percentage of tasks that are accepted by edge devices. One drawback of their deep reinforcement model is that all the decisions are made by a central server and the states in the trained deep reinforcement model also change size relative to the number of edge devices in the system. This makes the system inflexible and unscalable, as the system cannot have too many edge devices and the model should be retrained each time an edge device is added or removed.

In [8], there are three types of devices: wireless users (WU), U-MEC, and F-MEC. In their model, tasks arrive at WUs, which are responsible for making a decision to offload the task to a U-MEC or a F-MEC. U-MECs are unmanned vehicle-assisted servers, which have limited power and processing capacity but are generally closer to WUs and can move around closer to WUs. On the other hand, F-MECs have a fixed location, but no power restrictions and more processing power. For the optimization criteria, they used a combination of total delay, energy consumption of WUs, and bandwidth of MEC devices.

For the optimization model, they used SARSA [9] and dueling deep Q-learning [10] models. SARSA is a reinforcement learning method in which the model is updated after each step when a reward for a task is obtained. In a dueling deep Q-learning model, two networks are used to estimate the value of the current state and the advantage of a chosen action. They show that, generally, the dueling deep Q-learning model performs better than SARSA.

Zhang et al. [11] used a game theory model for the task offloading problem. They considered a model with one fixed MEC and one flying MEC as edge devices to serve mobile users on the ground. In their model, all mobile users are connected to both fixed and flying MECs and a decision to offload a task is made collectively. At each step, each mobile device advertises the best decision it can take to minimize a linear combination of the delay and power consumption costs for the current task. After all mobile devices have made their advertisements, the mobile device with action for which the total cost function is minimized "wins" and its decision is recorded on a list. This process is repeated until the list does not change. Their report, however, did not take into account the delay introduced by the execution of this game theoretic algorithm, as it can take several repetitions until the list comes to an equilibrium. Additionally, it does not explain how tasks will be treated if they do not arrive at the same time on all mobile devices.

In [12], the authors introduce the concept of a candidate network. In this model, instead of training only one DQL network, $n$ DQL networks are trained, where some are trained after every $C$ iterations while others are trained after every iteration. At each step, the network with the smallest $Q$ value is selected as the candidate network to be used for action selection. Regarding the optimization criteria, it uses a combination of power consumption, cost, and load balancing. The cost is a dynamic function of the amount of processing resources remaining at an edge device at the moment.

In [13], each mobile device can connect to multiple edge devices and when a task is offloaded to an edge device, it is either executed or dropped. Each task has a deadline by which it needs to be fully executed; if an edge device does not have the resources to execute a task on time, the task will be dropped. With regard to the cost function, if a task is executed, the cost function will be a function of the time delay, and if a task is dropped, a constant cost value is considered. The optimization problem is to minimize the total cost of all tasks. To do this, a deep Q-learning model with LSTM [14] as mid-layer is employed. In this model, mobile devices choose an edge device to train a neural network instead of training it themselves. The reason for this is to alleviate the load on mobile devices, as edge devices have more processing power. However, a separate model is trained on the edge device for each mobile device using the experiences of that mobile device. This is a major disadvantage, as the edge device will need to train a relatively large number of neural networks which can use significant processing resources of the edge device. Additionally, if the mobile device is disconnected from the chosen edge device, a new neural network on another edge device must be trained and the previous experiences will be lost.

The authors of [15] also employed LTSM [14] mid-layers within their deep Q-network, which allows the network to memorize more information and reduce training time. They, however, introduced three types of MECs for task offloading: base stations (BS) with more processing power, but which are usually farther from mobile devices, road side units that have less processing power but are usually closer to mobile devices, and parked idle vehicles, which can be thought of as other mobile devices not currently in use. This architecture allows the system to take advantage of the mobile devices with no workload to assist other mobile devices. Similarly, the authors in [16] introduced a pairing scheme to pair resource-constrained mobile devices with idle mobile devices to balance the load between them. They, however, built their model with the assumption that the number of idle devices is always greater than the number of resource-constrained devices, which is not always the case.

In [17], mobile devices do not perform local execution and it is assumed that each mobile device is within the coverage of one and only one unmanned aerial vehicle (UAV), to which it offloads its tasks. This requires that the area of coverage for UAVs not overlap.

Additionally, UAVs may be connected to a number of edge clouds (ECs) to further offload some of the tasks received from mobile devices. In a sense, in this system, UAVs can be thought of as traditional mobile devices with limited processing and energy capacity, and ECs as traditional MECs. Even though each UAV trains its own model for task offloading, a multi-agent TD3 algorithm [18] is employed to promote cooperation between UAVs instead of each UAV maximizing its own reward. On the contrary, Chen et al. in [19] argue that it is fairer for each mobile device to minimize the workload and energy consumption for itself, as mobile devices are owned by different users. Hence, they introduce a multi-agent deep deterministic policy gradient (MADDPG) [20] framework to optimize in such a cooperative–competitive environment.

## 3. System Model

In this model, we have $n$ mobile devices $M = \{m_1, m_2, ..., m_n\}$ and $m$ edge nodes $E = \{e_1, e_2, ..., e_m\}$. Mobile devices and edge nodes can have wireless connections to each other. Additionally, tasks can arrive at any time in mobile devices and should be computed either in the mobile device or offloaded to an edge node. In the following subsections, the task model, mobile devices, and edge nodes are described in more detail.

### 3.1. Task Model

Tasks can arrive at mobile devices at any time. In the simulation presented in this work, task arrivals follow a Poisson process, where the exact task arrivals are random but the average rate of task arrival is constant at $\lambda$. In our simulation, time is continuous and is not discretized into time slots. This allows us to accurately simulate a Poisson process, which, in turn, more accurately simulates the real-world conditions. In our model, task offloading decisions are binary, which means each task must be fully executed by a mobile device or an edge device. A task cannot be partially executed by both mobile and edge devices.

Let T be the set of all arrived tasks in our system $T = \{t_1, t_2, ...\}$. At the beginning, the set $T$ is empty, and as time passes, more tasks are added into the set. Let $m(t_i) = m_j$ indicate the mobile node $m_j$ at which task $t_i$ has arrived. Additionally, let $l(t_i) = l$ and $d(t_i) = d$ indicate the size $l$ and computational workload $d$ of the task $t_i$, respectively. The computational workload of a task indicates the number of floating point operations required to execute the corresponding task. In this work, we use floating point operations instead of CPU cycles to indicate the workload of tasks. Flops is a unit of measurement that stands for floating-point operations per second. This measurement is more accurate than CPU cycles for tasks that require many floating point operations. For a task to be executed in one second, a processor with a processing capacity of $d(t_i)$ flops is required. If a processor has higher processing capacity, the task will execute in less than one second, and if a processor has lower processing capacity, the task will run in more than one second.

### 3.2. Mobile Devices

When tasks arrive at mobile devices, the devices have the responsibility to execute the tasks as quickly as possible. In out model, each mobile device is connected to at most one edge node. Connecting to more edge nodes is not allowed here, as maintaining multiple connections incurs more power consumption for mobile nodes, which need to conserve their batteries. Additionally, it complicates decision-making for mobile nodes, which requires them to dedicate more power for task offloading decisions and in turn, consume more energy.

We define $e(m_i) = e_j$ to denote the edge device $e_j$ that the mobile device $m_i$ is connected to and $r(m_i) = r$ to denote the data rate $r$ of the connection between mobile device $m_i$ and its connected edge.

### 3.2.1. Task Offloading

After a task arrives at a mobile device, a decision is immediately made to either run the task locally or transmit it to an edge node for remote execution. Different algorithms or

models can be used to make this offloading decision. Later in this work, we discuss some possible algorithms and their corresponding performance. If a task $t_i$ is to be executed locally, it is then added to a local execution queue $q(m_i)$, that is, $q(m_i) \leftarrow q(m_i) \cup \{t_i\}$. Otherwise, if a task is to be transmitted to the connected edge, it is added to a transmission queue $t(m_i)$, which can be represented as $t(m_i) \leftarrow t(m_i) \cup \{t_i\}$.

### 3.2.2. Local Execution

The local execution queue is a first-in-first-out queue. That is, tasks arriving in the queue earlier will also be executed earlier. Each mobile device has $p(m_i)$ flops of processing power. As a result, given task $t_i$, the amount of time required to run the task locally is:

$$d_{l.e}(t_i) = d(t_i)/p(m_i) \tag{1}$$

Additionally, the amount of time required for the task to wait in the local execution queue is:

$$d_{l.eq}(m_k, t_i) = \alpha.d_{l.e}(t_c) + \sum_{t_j \in q(m_k)} d_{l.e}(t_j) \tag{2}$$

where $\alpha$ is the ratio of the currently executing task that is executed, $t_c$ is the currently executing task, and $m_k$ is the corresponding mobile device. If no task is currently executing, the value of term $\alpha.d_{l.e}(t_c)$ will be zero.

As a result, if a task $t_i$ is to be executed locally, it will require a $d_{l.et}(t_i)$ amount of time:

$$d_{l.et}(m_k, t_i) = d_{l.e}(t_i) + d_{l.eq}(m_k, t_i) \tag{3}$$

The energy consumption of executing a task $t_i$ at mobile device $m_j$ is

$$e_e(m_j, t_i) = c_e(m_j).d(t_i) \tag{4}$$

where $c_e(m_j)$ is the energy consumption in joules for performing one floating point operation in the mobile device $m_j$.

### 3.2.3. Transmission

The transmission queue is also a first-in-first-out queue. Each task must wait until all earlier tasks are transmitted before it can be transmitted. In our simulation, the data rate for a connection is approximated using the Shannon–Hartley theorem (5).

$$r = B.\log_2(1 + P/N) \tag{5}$$

where $B$ is the bandwidth of the channel, $P$ is the received power of a transmission in watts, and $N$ is the received noise power in watts. The received power is approximated using Friis transmission equation

$$P = P_t.G_t.G_r.(\omega/(4\pi.D))^2 \tag{6}$$

where $P_t$ is the transmitted power of the transmitter in watts, $G_t$ and $G_r$ are the transmitter and receiver gains in dBi, $\omega$ is the wavelength of the channel in meters, and D is the distance between the transmitter and receiver in meters. The wavelength of the channel $\omega$ can be calculated using the frequency of the channel $f$.

$$\omega = c/f \tag{7}$$

where $c$ is the speed of light. To calculate the received noise power, we approximate the received noise with a Gaussian white noise $N_g$ in dBi; then, we convert it into watts.

$$N = 10^{N_g/10} \tag{8}$$

Let $r(m_i, e_j)$ be the data rate between mobile device $m_i$ and the edge device $e_j$ calculated using Equation (5). The amount of time required to transmit a task $t_k$ is:

$$d_{l.s}(m_i, t_k) = l(t_i)/r(m_i, e_j) \tag{9}$$

If there are tasks in the transmit queue, the arriving task must wait for them to be transmitted first; the time for the task to wait in the transmit queue $t(m_i)$ is:

$$d_{l.sq}(m_k, t_i) = \alpha.d_{l.s}(m_k, t_c) + \sum_{t_j \in t(m_k)} d_{l.s}(m_k, t_j) \tag{10}$$

where $\alpha$ is the percentage of successful transmission for the task currently being transmitted, $t_c$ is the task currently being transmitted, and $m_k$ is the corresponding mobile device. If no task is being transmitted $\alpha.d_{l.s}(m_k, t_c)$ will be set to zero. Consequently, it will take $d_{l.st}$ to transmit the task $t_i$:

$$d_{l.st}(m_k, t_i) = d_{l.sq}(m_k, t_i) + d_{l.s}(m_k, t_i) \tag{11}$$

The energy consumption of transmitting a task $t_i$ at mobile device $m_j$ is

$$e_s(m_j, t_i) = c_s(m_j).d_{l.s}(m_j, t_k) \tag{12}$$

where $c_s(m_j)$ is the energy consumption in joules for one second of data transmission in the mobile device $m_j$.

### 3.3. Edge Nodes

After a task arrives at an edge node, a decision is immediately made to either execute the task or further offload it to another edge node. Different approaches for this decision will be discussed later in this report. If a task $t_k$ must be executed locally, it will be put in the execution queue $q(e_i)$; if it must be further offloaded to another edge device $e_j$, it will be put in the transmission queue of the corresponding connection $t(e_i, e_j)$, which can be represented as $t(e_i, e_j) \leftarrow t(e_i, e_j) \cup \{t_k\}$.

#### 3.3.1. Edge Execution

Similar to mobile devices, tasks in the execution queue are executed in a first-in-first-out manner. Executing a task $t_i$ will take $d_{r.e}(t_i)$ amount of time:

$$d_{r.e}(t_i) = d(t_i)/p(e_i) \tag{13}$$

where $p(e_i)$ is the processing power of each core of an edge device. Edge nodes have more processing cores and they can run multiple tasks concurrently. Whenever a core has finished executing a task, it checks the execution queue and if there is a task waiting to be executed, it will pick up the task for execution. The wait time $d_{r.eq}$ for each task $t_i$ is:

$$d_{r.eq}(e_k, t_i) = \alpha.d_{r.e}(t_c) + \sum_{t_j \in q_c(e_k, t_j)} d_{r.e}(t_j) \tag{14}$$

where $\alpha$ and $t_c$ are the ration and the current task running at the corresponding core. If no task is currently running at this core, $\alpha.d_{r.e}(e_k, t_c)$ will be replaced with zero. $q_c$ is the set of tasks that will be running on the corresponding core. This set is defined as:

$$q_c(e_k, t_j) = \{t | t \in q(e_k), core(t_j) = core(t)\} \tag{15}$$

where $core(t_j)$ indicates the processing core at which task $t_j$ will be run in edge node $e_k$. The total wait time for executing task $t_j$ at the edge node $e_k$ is

$$d_{r.et}(e_k, t_i) = d_{r.e}(t_i) + d_{r.eq}(e_k, t_i) \tag{16}$$

To calculate the total wait time for a task $t_i$ from the moment it arrives at the mobile device $m_j$ until it is executed at the edge node $e_k$, provided that the task is sent from a mobile device to this edge device, we have

$$d_{r.o}(m_j, e_k, t_i) = d_{l.st}(m_j, t_i) + d_{r.et}(e_k, t_i) \tag{17}$$

### 3.3.2. Transmission

When a task arrives at an edge node, it can be further offloaded to another edge node if it is was not received from an edge node. This limitation is in place for the simplicity of the network. Edge nodes, however, can maintain connections to multiple other edge nodes at the same time. In our simulation, the data rate between each two connected edge nodes $r(e_i, e_j)$ is calculated using Equation (5). The time required for re-transmission of a task $t_i$ from its current edge $e_c$ to another edge node $e_k$ is

$$d_{r.s}(e_c, e_k, t_i) = l(t_i)/r(e_c, e_k) \tag{18}$$

and the amount of time the task $t_i$ must wait in the transmission queue $t(e_c, e_k)$ is

$$d_{r.sq}(e_c, e_k, t_i) = \alpha.d_{r.s}(e_c, e_k, t_c) + \sum_{t_j \in t(e_c, e_k)} d_{r.s}(e_c, e_k, t_j) \tag{19}$$

where $t_c$ is the current task being transmitted. If no task is being transmitted, $\alpha.d_{r.s}(e_c, e_k, t_c)$ should evaluate to zero. The total transmission time for task $t_i$ is

$$d_{r.st}(e_c, e_k, t_i) = d_{r.s}(e_c, e_k, t_i) + d_{r.sq}(e_c, e_k, t_i) \tag{20}$$

Consequently, the total time to run a task $t_i$ after transmission to a second edge node can be calculated as

$$d_{r.o2}(m_j, e_c, e_k, t_i) = d_{l.st}(m_j, t_i) + d_{r.st}(e_c, e_k, t_i) + d_{r.et}(e_k, t_i) \tag{21}$$

## 4. Problem Definition and Optimization Approaches

When a task arrives at a mobile device, it is the responsibility of the mobile device to determine whether the task should be executed locally or be sent to the connected edge device. Similarly, when a task arrives at an edge device, it is the responsibility of the edge device to decide whether to further offload the task. Ultimately, we want to make these decisions at mobile devices and edge devices so that the total time from the moment a task arrives at a mobile device to the moment the task is executed is minimized. Additionally, we want to minimize the energy consumption at mobile devices for processing and transmitting tasks.

We can formulate these requirements into the following minimization problem:

$$\begin{aligned} Minimize : \alpha. \sum_{t \in T} (c_1(t).d_{l.et}(m(t), t)+ \\ c_2(t).d_{r.o}(m(t), e(m(t)), t)+ \\ c_3(t).d_{r.o2}(m(t), e(m(t)), e_k(t), t))+ \\ \beta. \sum_{t \in T} (c_1(t).e_e(m(t), t) + (c_2(t) + c3(t)).e_s(m(t), t)) \end{aligned} \tag{22}$$

subject to the following constraints

$$c_1(t), c_2(t), c_3(t) \in \{0, 1\} \quad \forall t \in T \tag{23}$$
$$c_1(t) + c_2(t) + c_3(t) = 1 \quad \forall t \in T \tag{24}$$
$$e_k(t) \in c(e(m(t))) \quad \forall t \in T \tag{25}$$

where $c(e_i)$ is a set of edge nodes to which edge node $e_i$ has a connection. $\alpha$ and $\beta$ are arbitrary coefficients used to modify the importance of time delay and energy consumption with respect to each other.

In the proposed solutions, it is assumed that the workload of a task $d(t)$ is not provided to the algorithm that is making the offloading decisions. The reason for this restriction is that in the real world, the workload of a task is not available before a task is executed in most cases.

*4.1. Greedy Approach*

In the greedy approach, we try to minimize the cost of individual tasks with the aim to minimize the overall cost of Equation (22). Firstly, as shown in Algorithm 1, we define a function to approximate the workload of a task using the average observed workload of $M$ previously completed tasks. As is depicted in Algorithm 2, this procedure repeats until all tasks have arrived. This algorithm is applied to both mobile and edge devices, regardless of the type of device. In the algorithm, local will refer to the current device at which task has arrived and remote will refer to an edge device to which a task may be offloaded. At each repetition, if a task is arrived from the network and from an edge device, meaning the current device is also an edge, the task is simply inserted into the local execution queue. If a task is not arrived from an edge device, state information is retrieved from the current device, including the size of the local execution queue $(l_s)$, the total size of the tasks in the transmission queue $(s_s)$, the size of the current task $(l(t))$, and a cached value representing the total number of tasks in the remote device. To acquire this cached value, every edge device periodically sends the total number of tasks in its execution queue to every other device, including both edge and mobile devices. This mechanism is elaborated in Algorithm 3, which is also used in the deep Q-learning approach and is explained in more details in the corresponding section.

---

**Algorithm 1** Greedy algorithm—workload approximation

---

1:  $W_l \leftarrow$ an empty list $\qquad\qquad\qquad\qquad\qquad$ ▷ For storing task workloads
2:  **repeat**
3:  $\quad$ wait until a task result arrives
4:  $\quad$ $r \leftarrow$ newly arrived task result
5:  $\quad$ $w \leftarrow$ the workload of task $r$
6:  $\quad$ insert $w$ into $W_l$
7:  $\quad$ **while** the count of $W_l$ is higher than $M$ **do**
8:  $\quad\quad$ remove the oldest item in $W_l$
9:  $\quad$ **end while**
10:  $\quad$ $w_t \leftarrow average(W_l)$
11:  **until** all task results have arrived

---

Next, the delay and power consumption of local execution and transmission of the task is approximated. These approximations are based on the number of tasks and their size. Arguably, the difficulty in finding an accurate approximation to these values is one of the drawbacks of the greedy approach. Having calculated the delay and power consumption, the value $\alpha.d_l + \beta.p_l$ represents the cost of local execution and the value $\alpha.d_r + \beta.p_t$ represents the cost of offloading, where $d_l$, $d_r$, $p_l$, and $p_t$ are the delay for local and remote execution and power consumption for local execution and transmission, respectively. The values $\alpha$ and $\beta$ are set using similar variables in Equation (22). Finally, if the cost of local execution is less than offloading the task, the task is inserted into the local execution queue. Otherwise, the task is inserted into the transmission queue.

---

**Algorithm 2** Greedy algorithm

---

1: **repeat**
2:    wait until a task arrives
3:    $t \leftarrow$ newly arrived task
4:    **if** task arrived from an edge device **then**
5:        insert task $t$ into local execution queue
6:    **else**
7:        $w_t \leftarrow$ the average task workload obtained from the workload approximation algorithm
8:        $l_s \leftarrow$ retrieve local execution queue task count
9:        $s_s \leftarrow$ retrieve transmission queue total task size
10:       $s_c \leftarrow$ retrieve transmission queue task count
11:       $E_l \leftarrow$ local power consumption per floating point operation
12:       $E_t \leftarrow$ local power consumption per second of data transmission
13:       $f_l, f_r, r \leftarrow$ local flops, remote flops, connection data rate
14:       $p_l \leftarrow E_l.w_t$                ▷ approximating local execution power consumption
15:       $p_t \leftarrow E_t.l(t)/r$              ▷ approximating transmission power consumption
16:       $d_l = (l_s + 1).w_t/f_l$
17:       $d_r = (l(t) + s_s)/r + (r_c + s_c + 1).w_t/f_r$
18:       **if** $\alpha.d_l + \beta.p_l \geq \alpha.d_r + \beta.p_t$ **then**
19:           insert task $t$ into local execution queue
20:       **else**
21:           insert task $t$ into transmission queue
22:       **end if**
23:   **end if**
24: **until** all tasks have arrived

---

**Algorithm 3** Edge state update algorithm

---

1: **while** *true* **do**
2:    $q \leftarrow$ number of tasks in local queue
3:    $e \leftarrow$ the unique identifier of this edge
4:    **for all** *node* in connected nodes **do**
5:        transmit $(e, q)$ to *node*
6:    **end for**
7:    wait for $c$ seconds
8: **end while**

---

*4.2. Deep Q-Learning Approach*

To use Q-learning [21] for our problem, we should model our problem into a Markov decision process (MDP) [22]. An MDP has a finite or infinite set of states and actions where we take an action to go from one state to another. The new state after each action depends on the previous state and the action taken. However, new states are not deterministic and each action can result in many different states, though with different probabilities.

In a Q-learning algorithm, given an MDP, whenever we take an action to move from one state to another, we receive a reward based on how good that action was. The purpose of Q-learning is to choose an action from a set of possible actions at each state to maximize the sum of all rewards earned during the execution of the algorithm. Q-learning in its vanilla form uses a table to approximate the total reward that can be earned for every possible combination of states and actions if we use the Q-learning algorithm with this table thereafter. This approach, however, does not work if the state space is too large or continuous, as the length of the table will become extremely large or infinite. Modeling our problem with an MDP will result in an infinite number of states; hence the vanilla form of Q-learning cannot be employed. To overcome this issue, we can use a deep Q-learning model instead of keeping a table of records. In the following subsections, we explain

the details of Q-learning, modeling our problem into an MDP, and the deep Q-learning approach.

### 4.2.1. Q-Learning Algorithm

In the Q-learning algorithm [21], we keep a table to store a value for any state and action. A typical table can have columns corresponding to actions and rows corresponding to possible states, and each value approximates the total reward that can be earned for the state and action it is specifying. To train this table, whenever we take an action at a state, we update the Q-value in our table corresponding to that state and action. We define $(s_1, s_2, ...)$ to denote the sequence of states we will pass through during the execution of this algorithm. We also define $(a_1, a_2, ...)$ as the corresponding actions taken at each state. Hence, at each step, we move from state $s_t$ to state $s_{t+1}$ by taking action $a_t$. Additionally, the Q-table is updated for state $s_t$ and action $a_t$ using

$$Q(s_t, a_t) \leftarrow Q(s_t, a_t) + \alpha.\delta(Q, s_t, a_t) \tag{26}$$

where $\alpha$ is the learning rate and $\delta(Q, s_t, a_t)$ is the temporal difference [23] defined as

$$\delta(Q, s_t, a_t) = r + \gamma.\max_{a \in A} Q(s_{t+1}, a) - Q(s_t, a_t) \tag{27}$$

in which $r$ is the reward earned from moving from state $s_t$ to $s_{t+1}$ by taking action $a_t$, $\gamma$ is the discount factor, and $A$ is the set of all possible actions. Both $r$ and $\gamma$ are values between zero and one. The learning rate affects how quickly the table will converge. If the learning rate is too small, the table might take a long time to converge, and if the learning rate is too large, the table might never converge, as it might jump over the correct values. The discount factor indicates how much we value earlier rewards. That is, if the discount factor is closer to zero, we value the immediate reward more, and if the discount factor is closer to one, the long-term reward is valued more over immediate reward.

### 4.2.2. Modeling the Markov Decision Process

As a decision on a newly arriving task must be taken both in a mobile device and an edge node, we design two MDPs corresponding to each. To model an MDP, we need to define the states and actions. Regarding the actions, in both cases, we have two possible actions: executing a task locally or transmitting it to an edge node for execution. With regard to states, for a mobile device, the states are designated as

$$(a_s, d_r, l_q, t_s, t_q, r_q) \tag{28}$$

where $a_s$ is the arriving task size, $d_r$ is the average data rate of the connection to the connected edge device, $l_q$ is the number of tasks in the local execution queue, and $t_s$ and $t_q$ are the total size and total number of the tasks in the transmission queue, respectively. $r_q$ is the total number of tasks in the connected edge node. For an edge device, states are designed as

$$(a_t, a_s, d_r, l_q, t_s, t_q, r_q) \tag{29}$$

where $a_t$ is the duration of time that has passed from the moment the task originally arrived at a mobile device until the current moment, when we are making the offloading decision. Other values are defined similarly to the values in the state of a mobile device. If an edge is connected to multiple edge devices, the time for the execution of the task at each edge is approximated and only the edge with the smallest time is used to complete the state. The approximation is calculated using

$$d_{r.st}(e_c, e_k, t_i) + d_{r.et}(e_k, t_i) \tag{30}$$

where $e_c$ is the edge node making the decision, $e_k$ is a connected edge for which we are approximating the task execution time, and $t_i$ is the current task. The main reason for using

this approximation and not calculating the Q-value for every connected edge is to avoid favoring the offloading of a task compared to local execution. To understand the reason for this behaviors, let us assume that the Q-value has the probability of $\alpha$ to incorrectly give a higher value for the action of transmission instead of local execution. If an edge is connected to $n$ other edge devices and we calculate the Q-value of all $n$ edge devices, the probability of receiving at least one incorrectly higher value to transmission will be

$$P(a > 0) = 1 - (1 - \alpha)^n \tag{31}$$

where $a$ is the number of connected edges for which the Q-value is incorrectly higher for transmission. This value is clearly higher than $\alpha$, which proves our notion that calculating multiple Q-values will favor transmission of a task.

### 4.2.3. Deep Q-Learning Algorithm

The domain of states in our model for mobile and edge devices are $\mathbb{N}^6$ and $\mathbb{N}^7$, respectively, and some of the values such as $l_w$ can become relatively large. As a result, using a Q-table to approximate the total reward for each state $s_t$ and action $a_t$ is inefficient, taking a huge amount of time for the table to converge. This, in turn, renders this method unusable, as our model should quickly adapt to changes in a changing environment. To overcome this issue, we can use an artificial neural network (ANN) to replace our Q-table. That is, we feed a state $s_t$ and an action $a_t$ as the input of our ANN and receive one real value as the output, which is our Q-value $Q_\theta(s_t, a_t)$. The parameters in our ANN collectively are denoted by $\theta$. To train our ANN, we define the loss function as [6]

$$loss(\theta, \theta') = \mathbb{E}_{(s_t, a_t, s_{t+1}, r) \sim U(T_b)}[(r + \gamma \cdot \max_{a \in A} Q_{\theta'}(s_{t+1}, a) - Q_\theta(s_t, a_t))^2] \tag{32}$$

where $\theta$ is the current value of our ANN parameters and $\theta'$ is a snapshot of our ANN parameters updated after every $n$ number of trainings.

ANNs are known to be difficult to converge when used as the Q-function; hence, the deep Q-learning model employs two techniques to alleviate this issue [6]. The first technique is to break the correlation between consecutive transitions by randomly sampling from a pool of transitions. This method is known as experience replay. The second technique is to use a snapshot of network parameters $\theta'$ when calculating the loss function, which should reduce the correlation with the current state of network parameters $\theta$.

With regard to the MDP, to calculate the state of a mobile or edge device, we need information about the workload and task counts at the edges to which the node has connections. To satisfy this requirement, each edge device periodically sends its workload and task counts to all the devices to which it has a connection. Algorithm 3 illustrates this process.

The algorithm repeats for the lifetime of the network. At each iteration, first, it retrieves $e$ and $q$, which are the unique identifier and task count in the execution queue of the corresponding edge device. Then, it sends this pair $(e, q)$ to every node it has a connection to. Lastly, it waits for $c$ seconds before repeating the process. Here, $c$ is an arbitrary constant; if the network has a high data rate, $c$ can be very small, and if the data rate is low, $c$ can have a larger value. When $c$ has a smaller value, mobile and edge devices will have a more recent view of the state of the network and will be able to make better decisions.

The procedure for when a task arrives at a mobile device is depicted in Algorithm 4. The procedure is repeated until all tasks have arrived. When a task arrives, it is stored in variable $t$.

---

**Algorithm 4** Task arrival algorithm for mobile devices

---

  1: **repeat**
  2:     wait until a task arrives
  3:     $t \leftarrow$ newly arrived task
  4:     $s \leftarrow$ retrieve current state updated with task $t$
  5:     $x \sim U(0,1)$
  6:     **if** $x < \epsilon$ **then**
  7:         $a \leftarrow$ randomly choose from $\{0,1\}$
  8:     **else**
  9:         $a \leftarrow argmax_a(Q_\theta(s,a))$ where $a \in \{0,1\}$
10:     **end if**
11:     **if** $a = 0$ **then**
12:         insert task $t$ into local execution queue
13:     **else**
14:         insert task $t$ into transmission queue
15:     **end if**
16:     $s' \leftarrow$ retrieve current state
17:     store $(t, s, a, s')$ into partial transition buffer
18: **until** all tasks have arrived

---

Next, an updated state of the mobile device, illustrated in Equation (28), is recorded using the arrived task into variable $s$. Information received from the connected edge device using Algorithm 3 is used to calculate the value for $r_q$. Additionally, the values of $l_q$ and $t_q$ are increased by one. This means we assume that the task is inserted into both local and transmission queues and we want to remove the task from one of the queues by taking an action. This way of looking at this problem is necessary for the Q-learning algorithm to work properly, as we expect $\max_{a \in A} Q(s_{t+1}, a)$ to have a smaller value compared to $Q(s_t, a_t)$. In other words, when a task arrives, the workload of the task should be reflected in the state of the mobile device, which is satisfied by adding it to both the local and transmission queues.

A random value $x$ is then drawn using uniform distribution $U(0,1)$. If $x$ is smaller than $\epsilon$, we randomly choose an action. Hence, $\epsilon$ is used to control the rate of exploration/exploitation of our model. If $x$ is not smaller than $\epsilon$, then the Q-value for both local execution ($a = 0$) and transmission ($a = 1$) are calculated using the current parameters $\theta$ for our ANN, and the value of $a$ for which the Q-value is higher is returned. Based on the chosen value for $a$, task $t$ is inserted into the local or transmission queues. After the insertion, the state of the mobile device is recorded into $s'$, this time without any modification. Finally, the transition $(t, s, a, s')$ is recorded into the partial transition buffer. This transition from $s$ to $s'$ is considered partial because we still do not know the reward that will be obtained for the action $a$. This reward will arrive when the task is executed and the task result arrives at the mobile device.

As depicted by Algorithm 5, the procedure for when a task is received by an edge device is almost identical to that of a mobile device. The only difference is that when a task is received, first, we examine to determine if the task was received from an edge device, that is, if the task was further offloaded by an edge device. If this is the case, we simply insert the task into the execution queue and no record for this transition is put into the partial transition buffer. Otherwise, an identical procedure to that of the mobile device is employed.

---

**Algorithm 5** Task arrival algorithm for edge devices

---

1: **repeat**
2:     wait until a task is received
3:     $t \leftarrow$ newly arrived task
4:     **if** $t$ is received from an edge device **then**
5:         insert task $t$ into local execution queue
6:     **else**
7:         $s \leftarrow$ retrieve current state modified with task $t$
8:         $p \leftarrow$ a uniform random value in $[0, 1]$
9:         **if** $p < \epsilon$ **then**
10:             $a \leftarrow$ randomly choose from $\{0, 1\}$
11:         **else**
12:             $a \leftarrow argmax_a(Q_\theta(s, a))$ where $a \in \{0, 1\}$
13:         **end if**
14:         **if** $a = 0$ **then**
15:             insert task $t$ into local execution queue
16:         **else**
17:             insert task $t$ into transmission queue
18:         **end if**
19:         $s' \leftarrow$ retrieve current state
20:         store $(t, s, a, s')$ into partial transition buffer
21:     **end if**
22: **until** all tasks have arrived

---

Algorithm 6 illustrates the procedure when a task is executed either on an edge device or a mobile device. If a task is executed on an edge device, this procedure is executed first on the edge device and again when the task is received by the mobile device. Firstly, the received result is stored in variable $a$. If no item exists in this device's partial transition buffer for $a$, the rest of the procedure is ignored and the algorithm waits for the next task result to arrive. On the other hand, if an item corresponding to $a$ exists in this device's partial transition buffer, the item is retrieved, stored in $(t, s, a, s')$, and removed from the partial transition buffer. Next, a reward value $r$ is calculated for task $t$ using the received result $a$, and a completed transition $(t, s, a, s', r)$ is generated. If this device is a mobile device, then the completed transition is sent to the connected edge device for training. On the other hand, if this device is an edge device, first, the completed transition is added to the edge transition arrival queue so that it will be used for training the Q-learning algorithm. Secondly, the completed transition is also sent to nearby edge devices to help them train their own Q-learning models.

---

**Algorithm 6** Transition completion algorithm for mobile and edge devices

---

1: **repeat**
2:     wait until a task result arrives
3:     $a \leftarrow$ receive task result
4:     **if** partial transition for $a$ exists **then**
5:         $(t, s, a, s') \leftarrow$ retrieve partial transition for $a$
6:         $r \leftarrow$ calculate reward for $a$
7:         **if** this device is a mobile device **then**
8:             transmit completed transition $(t, s, a, s', r)$ to connected edge device.
9:         **else** (this device is an edge device)
10:             send $(t, s, a, s', r)$ to edge transition arrival queue
11:             transmit completed transition $(t, s, a, s', r)$ to connected edge devices.
12:         **end if**
13:     **end if**
14: **until** all task results have arrived

---

As a result, each edge device trains two Q-learning models: one for the connected mobile devices and one for making decisions itself. With regard to connected mobile devices, it receives transitions from all connected mobile devices and uses those transitions to train a Q-learning model. Each edge device also transmits all the transitions that it has received from mobile devices to connected edge devices, so that those edge devices can also use those experiences. Hence, each edge device uses the experiences of the connected mobile devices and the connected edge devices. This process is elaborated in Algorithm 7. With regard to the Q-learning model for edge-level decision-making, experiences from both the edge device and all the connected edge devices are used to train the Q-learning model. To achieve this, whenever a transition is arrived at the transition arrival queue, it is added to the transition buffer of the edge device for the purpose of training the model. The procedure is explained in Algorithm 8.

---

**Algorithm 7** Q-training algorithm in edge devices for connected mobile devices

---

1: $i \leftarrow 0$
2: **repeat**
3:     wait until a completed transition arrives
4:     $(t, s, a, s', r) \leftarrow$ receive transition
5:     **if** transition was arrived from a mobile device **then**
6:         transmit $(t, s, a, s', r)$ to connected edge devices
7:     **end if**
8:     store $(t, s, a, s', r)$ in transition buffer $T_{mb}$
9:     **if** $size(T_{mb}) > N$ **then**
10:        remove oldest item from transition buffer $T_{mb}$
11:     **end if**
12:     $i \leftarrow i + 1$
13:     **if** $i \geq n$ **then**
14:        $S \leftarrow$ sample $m$ items from transition buffer $T_{mb}$
15:        $train(Q_\theta^m, S)$
16:        $i \leftarrow 0$
17:     **end if**
18: **until** all task results have arrived

---

**Algorithm 8** Q-training algorithm in edge devices for edge level decision making

---

1: $i \leftarrow 0$
2: **repeat**
3:     wait until a completed transition arrives in edge transition queue
4:     $(t, s, a, s', r) \leftarrow$ receive transition
5:     store $(t, s, a, s', r)$ in transition buffer $T_{eb}$
6:     **if** $size(T_{eb}) > N$ **then**
7:        remove oldest item from transition buffer $T_{eb}$
8:     **end if**
9:     $i \leftarrow i + 1$
10:     **if** $i \geq n$ **then**
11:        $S \leftarrow$ sample $m$ items from transition buffer $T_{eb}$
12:        $train(Q_\theta^e, S)$
13:        $i \leftarrow 0$
14:     **end if**
15: **until** all task results have arrived

---

For both models, after every $n$ new insertions into the transition buffer $T_b$, a sample $S$ of size $m$ is drawn uniformly at random from the transition buffer $T_b$. This sample is then used to train the ANN using the loss function in Equation (32). This is commonly known as experience replay [6], as we are training our ANN using past experiences instead of training our ANN after each task result is received.

Lastly, as the procedure in Algorithm 9 shows, after every $c$ seconds, the edge device sends the current parameters of the ANN model for the mobile devices to all its connected mobile devices. When the parameters arrive, mobile devices will use them for task-offloading decision-making.

---

**Algorithm 9** Mobile ANN update algorithm

---

1: **while** *true* **do**
2:     $\theta \leftarrow$ network parameters from $Q_\theta^m$
3:     **for all** *node* in connected mobile nodes **do**
4:         transmit $\theta$ to *node*
5:     **end for**
6:     wait for $c$ seconds
7: **end while**

---

### 4.3. Routing Task Results after Disconnections

Before tasks are transmitted for offloading, a mobile node identifier value is packaged with the task for transmission. When the task arrives at an edge device, if the task is to be further offloaded, the node identifier of the edge device is also attached to the package with the task for transmission. These identifiers can be used to trace back the origin of a task in normal circumstances without disconnections.

In an event when a mobile device disconnects from an edge device and connects to a new edge device, the mobile device puts its node identifier inside a list structure and broadcasts a package containing the list into the network. For the sake of simplicity, we call the package sent with this list structure containing the node identifiers a "discovery package". Each time an edge device receives a discovery package, it checks the node identifiers inside the list. If the node identifier of the edge device is already inside the list, the package is dropped. On the other hand, if the list does not contain the node identifier of the edge device, the edge device adds its node identifier to the end of the list and rebroadcasts the package. In the latter case, the edge device also stores the discovery package in memory for later use.

When task execution is completed in an edge device, firstly the sequence of node identifiers augmented with the task are used to send back the task result to the original sender of the task. If the sequence requires transmission of the task results to a node to which the current node does not have a connection, then the current node searches its stored discovery packages to find another route to the mobile device. If no such route is found, the task result is stored in a waiting list and reattempted after each discovery package arrival. This process is repeated until the task result is delivered to the mobile device.

## 5. Results

A simulation was implemented and run for six task offloading approaches: greedy, deep Q-learning, reinforce [24], random, all tasks executed locally, and all tasks offloaded. For the greedy, reinforce, and deep Q-learning algorithms, both mobile and edge devices use the same method. However, for the random, local, and remote algorithms, edge devices always run all tasks locally without further offloading a task to another edge device. In the following subsections, the model for the simulation and the results are explained.

### 5.1. Simulation Model

This simulation consisted of four edge devices and a varying number of mobile devices. The environment for the simulation was a 90 m by 240 m rectangle where edge devices are located at $(30, 30)$, $(30, 90)$, $(75, 150)$, and $(25, 210)$ coordinates. All mobile devices were positioned randomly with a uniform random distribution in a rectangle within the simulation environment. The simulation was repeated for two selection of rectangles for placing the mobile devices. First, mobile devices were concentrated in a rectangle of size 60 by 240 m in the bottom section of the simulation environment, as is depicted in

Figure 1(left). Second, mobile nodes were concentrated in a rectangle of size 60 by 60 positioned at the vertical bottom and horizontal center of the simulation environment, as is shown in Figure 1(right). For the purpose of simplicity, the first model Figure 1(left) is referred to as "uniform" and the second model (right) is referred to as "centered". The reason for the asymmetry in the positioning of the edge and mobile devices is to better represent real world scenarios and to examine the efficiency of the provided methods in those settings.

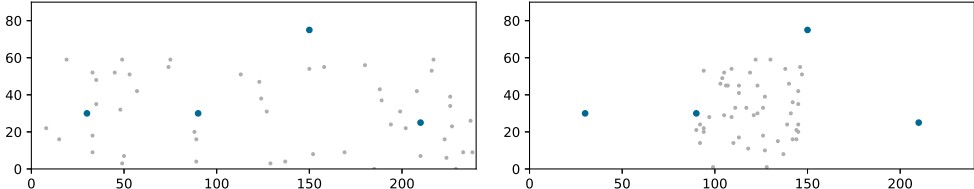

**Figure 1.** Two representations of the simulation environment where the distribution of mobile nodes is more uniform (**left**) and more centered (**right**).

After the simulation started, mobile nodes moved around the designated rectangle for mobile nodes (centered or uniform rectangles) inside the simulation environment using the random waypoint model. In the random waypoint model, each mobile node selects a random destination to move to and a random speed, and moves towards that destination in a straight line. After the node reaches its destination, it chooses another random location and repeats the process. Edge devices remained in place for the entire duration of the simulation.

The data rate between each pair of devices was modeled using Equation (5), with a random value of white noise used to calculate this value. The random value of white noise was calculated using a normal distribution. Additionally, the data rate for each task transmission was further randomized to a value in $[0.5, 1.5]$ of the calculated connection data rate using a uniform distribution. The general configuration values for the simulation are shown in Table 1.

**Table 1.** Simulation parameters.

| Parameter | Value |
| --- | --- |
| simulation duration | 300 s |
| default task arrival rate | 0.5 |
| node state transmission interval | 1 s |
| ANN parameters transmission interval | 1 s |
| min task size | 0.1 MBit |
| max task size | 1.0 MBit |
| min floating point operation per tasks | 0.25 trillion |
| max floating point operation per tasks | 1.5 trillion |
| mobile power consumption per tflops | 0.5 joules |
| mobile power consumption for transmission | 0.04 watts |
| mobile device antenna gain | 3 dBi |
| edge device antenna gain | 10 dBi |
| channel frequency | 2.4 GHz |
| channel bandwidth | 20 MHz |
| mean Gaussian white noise | −80 dBi |
| standard deviation Gaussian white noise | −10 dBi |
| mobile CPU core speed | 1 tflops |
| mobile CPU core count | 1 |
| edge CPU core speed | 5 tflops |
| edge CPU core count | 4 |
| learning rate | 0.001 |
| discount factor | 0.8 |
| training batch size | 200 |
| training interval | 5 |
| training buffer size | 10,000 |
| delay cost coefficient | 1 |
| power cost coefficient | 3 |
| simulation environment rectangle size | $80 \times 240$ m$^2$ |
| min mobile node velocity | 1 m/s |
| max mobile node velocity | 5 m/s |

In this simulation, tasks arrive using a Poisson process for the duration of the simulation. Before the simulation starts, a number of template tasks are created with task size and workload using a uniform distribution with the values in Table 1. During the execution of the simulation, each time a task needs to be created, first, it is sampled using one of the template tasks and then a small amount of noise is introduced in the task size and workload. This approach allows us to simulate the task conditions of many real world applications where similarly sized tasks have similar workloads. After a task arrives at a mobile device, it makes a decision to run the task locally or offload it to an edge device. A similar decision is made when an edge device receives a task from a mobile device. The greedy and deep Q-learning models are used to make such decisions in this simulation with varying values for task arrival rate and the number of mobile devices in our simulation. In the results subsection, the results for these different conditions are presented and evaluated.

### 5.2. Results

In the first set of experiments, we evaluated the performance of our proposed algorithm, DQL with edge training (DQL-edge), against five other methods: vanilla DQL where training is performed in local devices without experience sharing (DQL-local), reinforce using our proposed edge training architecture, greedy, local execution of all tasks, and randomly offloading or executing each task. It is worth mentioning that DQL-local imposes the burden of training the neural network on mobile devices; thus, comparing it with DQL-edge raises the question of how to account for this overhead. As the workload of training a neural network depends on hardware and software implementation, we neglect the training overhead in DQL-local and, as with all other methods, will only consider the workload for task execution. Even though this will give DQL-local an unfair advantage, we show that our proposed method can outperform vanilla DQL even if we neglect the training overhead. To compare the methods, tasks are binned in buckets of five seconds from the start to the end of the simulation and the average cost, delay, and power are calculated for each bin. For example, all the tasks arriving from $t = 0$ and $t = 5$ are placed in the same bucket and used to calculate the averages. We also carried out the experiment for 100 and 50 mobile nodes in the environment and with the mobile nodes using both the more uniform and centered locations. The results of the experiment are illustrated in Figure 2.

If the values for cost and delay increase overtime, it means that tasks are accumulating more and more in the execution queues of mobile or edge devices; hence, the delay and cost are getting worse over time. If this is the case, it means that the given method is unstable, as the delay will eventually increase and pass any acceptable range. As can be seen from the graphs, local execution of all tasks is always unstable, while the random and greedy methods manage to stay stable in some configurations. The DQL-edge and DQL-local methods on the other hand, managed to stay stable in all tested configurations, even though DQL-edge performed better in most configurations. This behavior can be attributed to the experience-sharing feature of DQL-edge, as the neural network will have more data to work with, which can lead to faster training and more resilience of the trained network. From the four tested configurations, 50 mobile nodes in the more uniform locations is the easiest configuration for a method to stay stable, as there are fewer mobile nodes to offload tasks to edge devices and the mobile nodes are distributed around all edge devices instead of a select few. In this configuration, all methods except for local execution remained stable and performed similarly. However, when the number of mobile nodes is 100 or when the mobile nodes are centered, the random method also fails to remain stable, while the greedy, DQL-local, and DQL-edge methods are still stable, even though the DQL-edge performs considerably better than the other methods. Lastly, when the mobile nodes are centered and we have 100 mobile nodes, only the DQL-local and DQL-edge methods manage to give acceptable results. The greedy method, while staying stable, only begins to plateau with a very high cost of around 50 units.

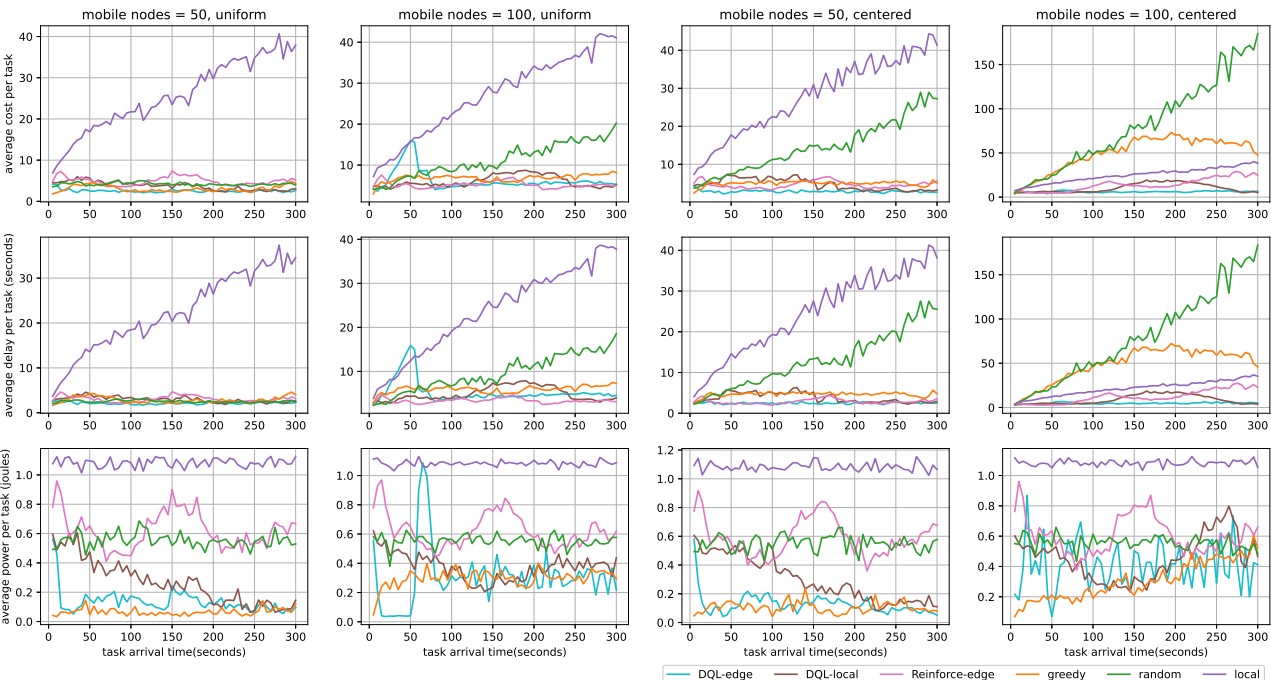

**Figure 2.** Comparing the cost, delay, and power consumption of the DQL-edge, DQL-local, reinforce-edge, greedy, random, and local task-offloading models for the duration of the simulation. The simulation was run for both centered and uniform positioning of mobile devices and when the number of mobile devices was 50 and 100.

The results for DQL-edge, DQL-local, reinforce-edge, and greedy are depicted in more detail in Figure 3. In this experiment, the simulation was run for mobile node numbers of 25, 50, 75, and 100, as well as for both uniform and centered configurations. As can be seen from the figure, DQL-local starts with a higher cost compared to the other two methods and closes the gap as time passes. This can be attributed to the slower learning speed of DQL-local, as it only has the experience of a single mobile device for training the network. We see some fluctuations with the reinforce method, which may be due to the fact that the reinforce method works best when we have long trajectories of experiences, which do not exist in our task offloading model. Overall, the DQL-edge method consistently yields better results compared to the DQL-local, reinforce, and greedy methods in all other tested configurations.

Next, we compare the mobile and edge processor utilization of our proposed method, DQL-edge, with DQL-local, Reinforce-edge, greedy, and locally executing all tasks. As can be seen from Figure 4, to minimize the power consumption of mobile devices, the algorithms offload most tasks to edge devices when there are abundant processing resources in edge devices, but when the resources on edge devices are constrained, they decide to execute more tasks locally. Even though all methods struggle to fully utilize edge processors in the centered with 100 mobile nodes configuration, DQL-edge manages to achieve an almost 90 percent edge processor utilization, which is higher than all other methods in that configuration.

Lastly, the cost values of an exhaustive experiment are presented in Table 2. These compare our proposed method, DQL-edge, with all other six methods, which are DQL-local, reinforce-edge, greedy, randomly choosing whether to offload a task (random), and all tasks executed locally (local) or remotely (remote). It also examines seven configurations of task arrival rates, $\lambda$, and mobile node counts in both centered and uniform settings. $\mu$ and $\sigma$ are the average cost and standard deviation of tasks for the entire duration of the simulation, and the graph underneath is the average cost as a function of time in the simulation. The graph, therefore, depicts the trend for cost as time passes. If the trend is upwards and the

standard deviation is high, it means that the model for the given configuration is unstable and unable to achieve a bounded cost if the simulation were to continue. Otherwise, the method is considered stable for the given configuration. As can be seen from the table, DQL-edge yields better results in almost all configurations when compared to other methods. When the mobile node count is low or the task arrival rate is low, this difference is minor, but when we increase either one, DQL-edge performs significantly better than any other method, especially in the centered configuration.

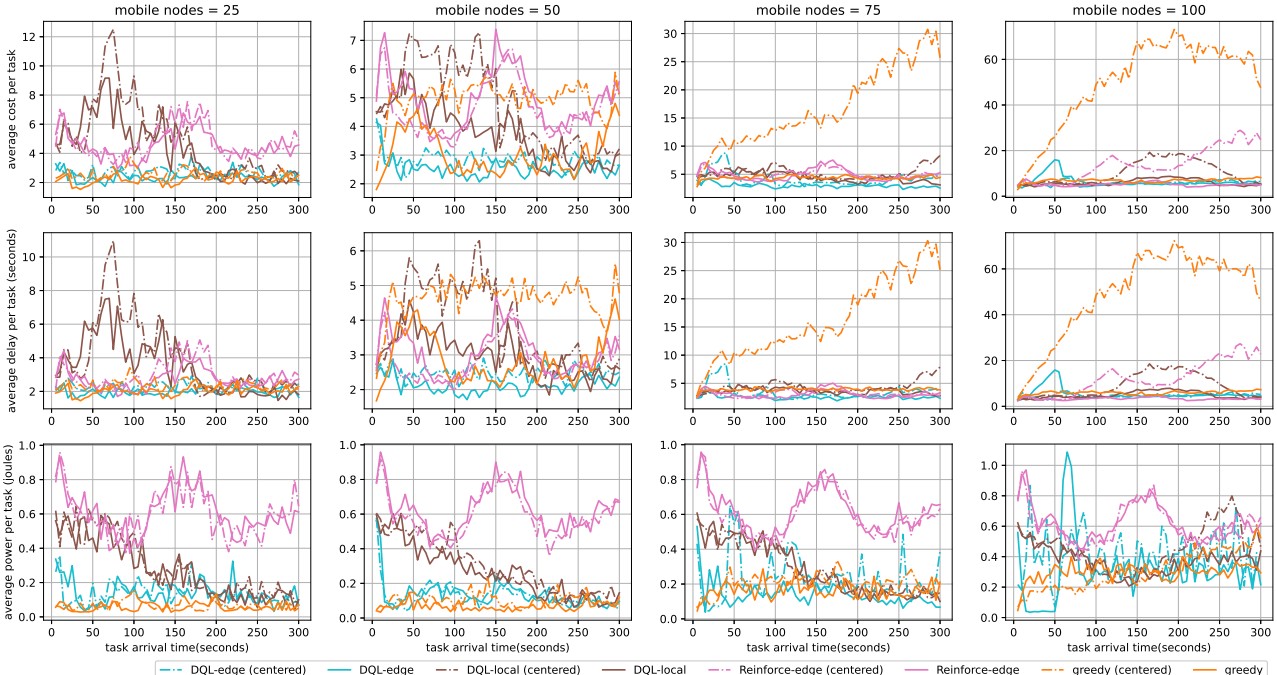

**Figure 3.** Comparing the cost, delay, and power consumption of the DQL-edge, DQL-local, reinforce-edge, and greedy task offloading models for the duration of the simulation. The simulation was run for both centered and uniform positioning of mobile devices and when the number of mobile devices was 25, 50, 75, and 100.

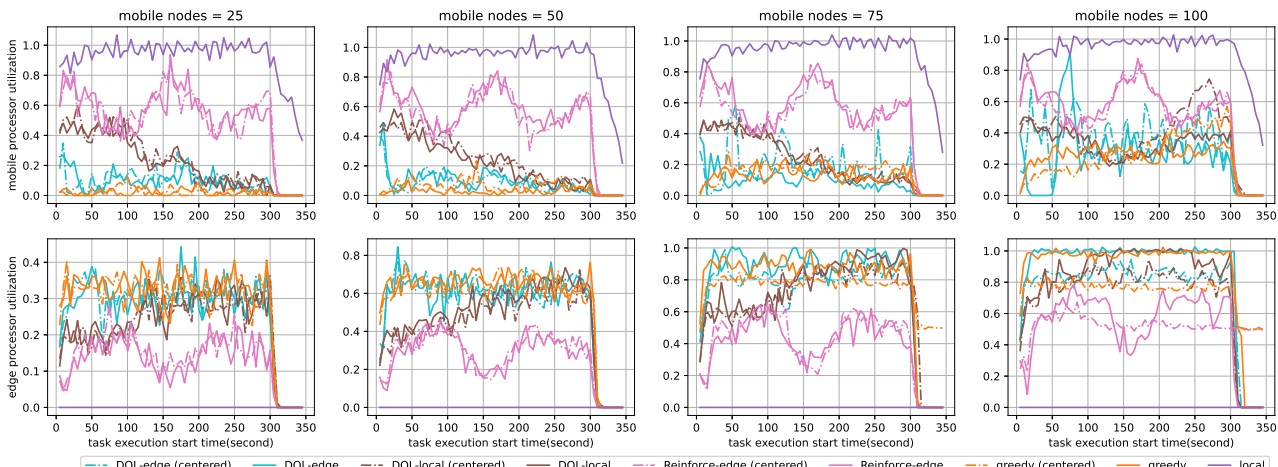

**Figure 4.** Comparing the mobile and edge processor utilization of the DQL-edge, DQL-local, reinforce-edge, greedy, and local task offloading models for the duration of the simulation with an extra 50 s after the task arrival process is stopped. The simulation was run for both centered and uniform positioning of mobile devices and when the number of mobile devices was 25, 50, 75, and 100.

**Table 2.** Mean and standard deviation values of cost and the cost trend over time for different task-offloading models, mobile device location model and counts, and task arrival. rates($\lambda$).

| | $\lambda$ = 0.50, count = 100 | $\lambda$ = 0.50, count = 75 | $\lambda$ = 0.50, count = 50 | $\lambda$ = 0.50, count = 25 | $\lambda$ = 0.25, count = 50 | $\lambda$ = 0.75, count = 50 | $\lambda$ = 1.00, count = 50 |
|---|---|---|---|---|---|---|---|
| DQL-edge (uniform) | $\mu = 6.26, \sigma = 3.26$ | $\mu = 2.99, \sigma = 1.93$ | $\mu = 2.54, \sigma = 1.84$ | $\mu = 2.39, \sigma = 1.80$ | $\mu = 1.99, \sigma = 1.56$ | $\mu = 3.52, \sigma = 2.31$ | $\mu = 7.27, \sigma = 2.87$ |
| DQL-edge (centered) | $\mu = 6.05, \sigma = 2.30$ | $\mu = 4.15, \sigma = 2.53$ | $\mu = 2.87, \sigma = 1.85$ | $\mu = 2.69, \sigma = 1.82$ | $\mu = 2.19, \sigma = 1.51$ | $\mu = 4.75, \sigma = 2.52$ | $\mu = 9.61, \sigma = 5.47$ |
| DQL-local (uniform) | $\mu = 6.04, \sigma = 4.29$ | $\mu = 4.47, \sigma = 4.38$ | $\mu = 3.78, \sigma = 4.08$ | $\mu = 4.36, \sigma = 6.27$ | $\mu = 3.15, \sigma = 2.57$ | $\mu = 6.04, \sigma = 8.85$ | $\mu = 15.87, \sigma = 17.41$ |
| DQL-local (centered) | $\mu = 10.40, \sigma = 7.94$ | $\mu = 5.24, \sigma = 4.80$ | $\mu = 4.72, \sigma = 5.61$ | $\mu = 5.00, \sigma = 7.17$ | $\mu = 3.25, \sigma = 2.46$ | $\mu = 9.48, \sigma = 8.65$ | $\mu = 22.67, \sigma = 17.89$ |
| Reinforce-edge (uniform) | $\mu = 5.21, \sigma = 3.46$ | $\mu = 4.96, \sigma = 3.63$ | $\mu = 4.96, \sigma = 3.76$ | $\mu = 4.58, \sigma = 3.29$ | $\mu = 3.99, \sigma = 2.57$ | $\mu = 8.00, \sigma = 7.70$ | $\mu = 32.80, \sigma = 38.25$ |
| Reinforce-edge (centered) | $\mu = 14.10, \sigma = 13.17$ | $\mu = 4.95, \sigma = 3.05$ | $\mu = 4.63, \sigma = 3.19$ | $\mu = 4.78, \sigma = 3.54$ | $\mu = 3.96, \sigma = 2.47$ | $\mu = 6.28, \sigma = 4.90$ | $\mu = 32.64, \sigma = 22.59$ |
| Greedy (uniform) | $\mu = 6.86, \sigma = 4.40$ | $\mu = 4.41, \sigma = 2.42$ | $\mu = 3.07, \sigma = 1.91$ | $\mu = 2.22, \sigma = 1.54$ | $\mu = 1.77, \sigma = 1.17$ | $\mu = 4.64, \sigma = 2.56$ | $\mu = 11.08, \sigma = 8.07$ |
| Greedy (centered) | $\mu = 50.18, \sigma = 50.98$ | $\mu = 16.98, \sigma = 14.25$ | $\mu = 4.97, \sigma = 2.19$ | $\mu = 2.59, \sigma = 1.62$ | $\mu = 2.01, \sigma = 1.18$ | $\mu = 16.52, \sigma = 12.50$ | $\mu = 39.77, \sigma = 39.15$ |
| Random (uniform) | $\mu = 11.27, \sigma = 14.64$ | $\mu = 4.43, \sigma = 2.97$ | $\mu = 4.20, \sigma = 3.06$ | $\mu = 4.33, \sigma = 3.24$ | $\mu = 3.78, \sigma = 2.57$ | $\mu = 6.18, \sigma = 5.88$ | $\mu = 20.90, \sigma = 19.25$ |
| Random (centered) | $\mu = 81.79, \sigma = 113.06$ | $\mu = 48.79, \sigma = 63.71$ | $\mu = 14.59, \sigma = 13.69$ | $\mu = 4.31, \sigma = 3.10$ | $\mu = 3.82, \sigma = 2.45$ | $\mu = 53.20, \sigma = 66.03$ | $\mu = 99.83, \sigma = 116.38$ |
| Remote (uniform) | $\mu = 115.45, \sigma = 132.65$ | $\mu = 60.48, \sigma = 78.97$ | $\mu = 18.44, \sigma = 21.47$ | $\mu = 2.45, \sigma = 2.05$ | $\mu = 1.80, \sigma = 1.23$ | $\mu = 72.36, \sigma = 90.23$ | $\mu = 123.69, \sigma = 140.89$ |
| Remote (centered) | $\mu = 453.74, \sigma = 344.45$ | $\mu = 323.52, \sigma = 237.81$ | $\mu = 163.05, \sigma = 124.42$ | $\mu = 15.27, \sigma = 12.00$ | $\mu = 22.28, \sigma = 13.94$ | $\mu = 336.57, \sigma = 240.81$ | $\mu = 477.35, \sigma = 347.67$ |
| Local | $\mu = 25.95, \sigma = 17.30$ | $\mu = 25.97, \sigma = 16.36$ | $\mu = 26.72, \sigma = 17.94$ | $\mu = 25.80, \sigma = 17.75$ | $\mu = 6.77, \sigma = 2.41$ | $\mu = 101.34, \sigma = 57.78$ | $\mu = 184.75, \sigma = 105.73$ |

## 6. Conclusions and Future Work

Even though a deep Q-learning model has been previously used in the literature for making the task-offloading decision, the computational burden of training an artificial neural network has often been neglected. To solve this issue, an artificial neural network was trained in edge devices and then was used for decision-making in all connected mobile devices. Not only does this alleviate the computational burden on mobile devices, but it also helps the neural network to converge significantly faster, as the experience of several mobile devices are used for training. In addition to the neural network trained in each edge device to be sent to connected mobile devices, a separate deep Q-learning model was trained in each edge device to enable the edge device to further offload some tasks to other edge devices. This mechanism helped edge devices to balance the load between each other and keep the overall delay small, as was demonstrated with the experiments containing centered mobile devices.

Finally, a routing solution was proposed to address the issue of moving mobile nodes, as a mobile nodes can move too far from their connected edge device and consequently become disconnected. This is a problem, as mobile devices can have offloaded tasks to the disconnected edge device while still waiting for the task results. The proposed routing solution addresses this problem by using broadcast packets to create a route from the disconnected edge device to the mobile device.

However, one of the limiting factors in choosing different deep reinforcement learning models for the issue of task offloading is that the experiences used to train a model are generated for each arrived task and, hence, they are not necessarily sequential events. That is, the environment can change drastically between the arrival of two tasks, this change is not necessarily a function of the actions taken by the model. As a result, each experience can be looked at as an experience independent to other experiences used for training, instead as a series of sequential events. Consequently, using reinforcement learning models

that take advantage of trajectories of states and actions is not possible without significant modifications to the modeling of the environment. We consider this issue as a topic for future works.

**Author Contributions:** Conceptualization, S.C. and A.Z.; methodology, A.Z.; software, A.Z.; validation, S.C. and A.Z.; formal analysis, A.Z.; investigation, S.C. and A.Z.; resources, S.C. and A.Z.; data curation, A.Z.; writing—original draft preparation, A.Z.; writing—review and editing, S.C. and A.Z.; visualization, A.Z.; supervision, S.C.; project administration, S.C. All authors have read and agreed to the published version of the manuscript.

**Funding:** This research received no external funding.

**Institutional Review Board Statement:** Not applicable.

**Informed Consent Statement:** Not applicable.

**Data Availability Statement:** All the implementations of the simulation and the generated results can be found at https://github.com/ahmadzendebudi/edge_simulation1 (accessed on 1 October 2022).

**Conflicts of Interest:** The authors declare no conflict of interest.

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
