# Peer review of "Designing a Deep Q-Learning Model with Edge-Level Training for Multi-Level Task Offloading in Edge Computing Networks"

_applsci, doi:10.3390/app122010664_

Round 1

Reviewer 1 Report

The paper presents a deep q-learning model to manage a task offloading decision process in mobile and edge devices. My comments are as follows:

- the authors should revise their work concerning the use of English.

- the authors should enhance the related work with additional models related to tasks offloading.

- the authors should compare their model with other model dealing with tasks offloading and not only with baseline models.

- if the algorithms adopted by mobile devices and edge nodes are the same the relevant discussion could be improved and eliminated double information.

Reviewer 2 Report

The proposed paper deals with a sophisticated simulation and benchmarking of different algorithms for the offloading of processing from mobile or IoT devices to edge devices, at fits the topics of the special issue.

While the core element of the paper (chap3, chpa4, chap5) are fine, other parts need a relevant review.

The introduction and chapter2 should be improved in terms of context definition, different relevant constraints to be considered, and SotA.

Starting from a cited reference, QoS evaluation should be considered as major point to drive further decisions, an acceptance criteria should be introduced, which indicates whether a task can be accepted by an edge device considering its QoS requirements; these aspects should be investigated more in detail.  Since the overall performance depends from latency, offloading time and processing time (including deep-learning training), a satisfaction time limit should be set, discarding all that exceed that limit. In this perspective, the cloud could be considered as an additional option, to be compared with all other options. In the result section it is mentioned that cost and delay increase overtime means that tasks are accumulating more and more in the execution queues of mobile or edge devices. These are the best candidate scenarios where cloud offloading would be worth considering it.

Also the final part needs a revision, first of all there is mention of cloud offloading while in the paper is not considered (and could be), so an effort to make the content more consistent is required. Then the case of load balancing between edge nodes should be better explained, trying to highlight as this could affect the final results. Some more details on the distribution of the computation of training neural network to support such kind of offloading algorithms should be better described.

Overall the paper is sound and interesting, written in a good english, just some typos, detailed proofreading suggested.

Referencies could be improved a bit.

Round 2

Reviewer 1 Report

The authors made updates in the paper, however, did not perfom a comparison with other relevant models providing some arguments for that. In my opinion, this comparison is necessary to expose the benefits of the approach. The allegations that the model is desgined for edge and cannot be compared with other schemes are not valid. Noone says that the authors should try to 'copy' other models especially those with a complex architecture but to 'borrow' the core models, apply them into their architecture and check the results.

I could indicate papers trying to do similar things, however, this not ethical from an Academic perspective for a Reviewer. Hence, it is the authors responsibility to check the papers referred in the respective literature and provide the comparative assessment.
